# Comedications with Immune Checkpoint Inhibitors: Involvement of the Microbiota, Impact on Efficacy and Practical Implications

**DOI:** 10.3390/cancers15082276

**Published:** 2023-04-13

**Authors:** Julien Colard-Thomas, Quentin Dominique Thomas, Marie Viala

**Affiliations:** 1Department of Medical Oncology, Montpellier Cancer Institute (ICM), University of Montpellier (UM), 34090 Montpellier, France; 2Oncogenic Pathways in Lung Cancer, Montpellier Cancer Research Institute (IRCM) INSERM U1194, University of Montpellier (UM), 34090 Montpellier, France

**Keywords:** immune checkpoint inhibitors, microbiota, comedications, corticosteroids, antibiotics, proton pump inhibitors

## Abstract

**Simple Summary:**

Since the approval of immune checkpoint inhibitors for solid tumors, concerns have been raised about the role of the microbiota on immunotherapy success and the potential negative impact of several drugs on oncological response and survival. In this review, we analyzed the existing data regarding such negative but also positive effect of many drugs on immunotherapy results. Most deleterious drugs concerning immunotherapy efficacy are corticosteroids, antibiotics and proton pump inhibitors, but preclinical as well as clinical studies have been conducted on other molecules such as metformin, statins, aspirin or beta blockers. The aim of this review is to elaborate a practical overview of current scientific data to help clinicians make critical treatment decisions by taking into account the totality of their patients’ comedications.

**Abstract:**

Immune checkpoint inhibitors (ICIs) have been a major breakthrough in solid oncology over the past decade. The immune system and the gut microbiota are involved in their complex mechanisms of action. However, drug interactions have been suspected of disrupting the fine equilibrium necessary for optimal ICI efficacy. Thus, clinicians are facing a great deal of sometimes contradictory information on comedications with ICIs and must at times oppose conflicting objectives between oncological response and comorbidities or complications. We compiled in this review published data on the role of the microbiota in ICI efficacy and the impact of comedications. We found mostly concordant results on detrimental action of concurrent corticosteroids, antibiotics, and proton pump inhibitors. The timeframe seems to be an important variable each time to preserve an initial immune priming at ICIs initiation. Other molecules have been associated with improved or impaired ICIs outcomes in pre-clinical models with discordant conclusions in retrospective clinical studies. We gathered the results of the main studies concerning metformin, aspirin, and non-steroidal anti-inflammatory drugs, beta blockers, renin-angiotensin-aldosterone system inhibitors, opioids, and statins. In conclusion, one should always assess the necessity of concomitant treatment according to evidence-based recommendations and discuss the possibility of postponing ICI initiation or switching strategies to preserve the critical window.

## 1. Introduction

The advent of immune checkpoint inhibitors (ICIs) in the 2010s has revolutionized cancer therapy. These molecules act very differently from previous oncological treatments and have become a standard of care for many cancers in metastatic or neoadjuvant/adjuvant settings (e.g., melanoma or lung carcinoma) [1,2]. Their mechanism of action is based on blocking the recognition of the lymphocyte ligand/receptor checkpoint pair, which activates and stimulates the immune system against tumor cells [3]. The main targets of these treatments for solid tumors are the programmed cell death protein 1 (PD-1), its ligand (PD-L1), and the cytotoxic T-lymphocyte-associated protein 4 (CTLA-4). However, not all patients benefit from the efficacy of these molecules. Primary resistance is often observed, as well as secondary resistance after an initial response. Elucidating some of the underlying mechanisms of resistance could help clinicians optimize treatment strategies. Because of the inherently pro-inflammatory therapeutic effect of ICIs, many classes of drugs that might impact the immune system may decrease their efficacy, most notably corticosteroids [4].

More recently, the important role of microbiota, mainly though not only intestinal, has been demonstrated on the antitumor activity of ICIs. As a result, it has been suggested that other therapeutic families known to induce dysbiosis, such as antibiotics or proton pump inhibitors, might reduce the effectiveness of ICIs [5,6]. In addition, other molecules have been associated in vitro and in animal models with increased or decreased efficacy of ICIs through other inflammatory or metabolic pathways, but it was not straightforward to transpose these results for clinical practice, and they need to be confirmed.

The main objective of this review is to draw a state of the art of the microbiota’s role in the efficacy of ICIs and to gather various studies that looked for an impact, positive or negative, on oncological outcomes under ICI treatment of the main therapeutic classes that might interact. We have focused on the three most commonly suggested drug families: corticosteroids, antibiotics, and proton pump inhibitors; and then we identified existing data for the other relevant molecules: metformin, aspirin and nonsteroidal anti-inflammatory drugs, beta blockers, renin-angiotensin-aldosterone system antagonists, opioids, and statins. 

## 2. Methodology of Literature Search and Selection

This review analyzes the existing data of the literature published between 2007 and 2023 related to the microbiota, its treatment-induced perturbations, and the impact of the microbiota and several comedications on ICI efficacy to give a broad overview on this increasingly important topic in clinical practice. In this regard, the authors searched well-known databases (PubMed, Cochrane, Google Scholar), using key words or combinations of them (i.e., immunotherapy; immune checkpoint inhibitors; immune checkpoint blockade; microbiota; immune system; corticosteroids; antibiotics; proton pump inhibitors; metformin; aspirin; non-steroidal anti-inflammatory drugs; beta blockers; renin-angiotensin-aldosterone system antagonists; angiotensin converting enzyme inhibitors; angiotensin-receptor blocker; opioids; statins). Most cited preclinical studies were selected, along with those providing striking results. Likewise, the most important meta-analyses were selected, from which retrospective studies were extracted. Those exploring only one drug and/or examining interesting patient subgroups were preferred. As a result, 143 references were cited as supporting the statements in this work.

## 3. The Microbiota and the Immune System

### 3.1. Microbiota: Definition and Composition

The microbiota is the collection of all the microorganisms that occupy a given environment, whereas the microbiome is defined as the set of all the genomic elements of a specific microbiota. The human microbiota living in and on human organisms is composed of thousands of species belonging to different kingdoms: bacteria, archaea, protozoa, fungi, animals, and viruses [7,8]. Dominant bacterial phyla include Firmicutes (30–50%), Bacteroidetes (20–40%), and Actinobacteria (3–10%). At the genus level, *Bacteroides* is the most abundant by far (up to almost 20%), followed by *Faecalibacterium* and *Bifidobacterium* [8]. The microbiota plays a fundamental role in the physiological functioning of the human body: among other things, it is involved in digestion, immune system development, and specific enzyme or vitamin synthesis. This is why the microbiome is sometimes considered as a second genome [7].

### 3.2. Relation with the Immune System

The role of gut microbiota on human host inflammation has been mostly studied in the context of inflammatory bowel diseases (IBD), where significant alterations in composition have been shown in affected patients compared to control samples [9]. A review by Pavel et al. reported the specifically increased and decreased bacterial genera in IBD patients and their consequences on mucosal injuries. They point out as well the beneficial potential of probiotics in the management of these diseases, particularly in combination. *Bifidobacterium* and *Lactobacillus* seem to be the most interesting strains in these pathologies. Results are encouraging with prebiotics and more variable with fecal transplantation [10,11]. Differences in gut flora composition have also been associated with pro-inflammatory status in relation to obesity, with the onset of allergies at school age, and with a high-risk human leukocyte antigen (HLA) genotype for celiac disease in children [12,13,14].

As illustrated in Figure 1, the microbiota’s important role in immune regulation involves, among other things, secreting of cytokines (e.g., interleukins, tumor-necrosis factor (TNF)-α) or matrix metalloproteinases, maintaining homeostasis, or producing T cells [15,16]. Its action has also been demonstrated in stimulating medullary and extra-medullary hematopoiesis for the development of certain components of the immune system, such as myeloid cell derivatives through granulocyte and/or monocyte progenitors differentiation [17].

### 3.3. Impact of Gut Microbiota on Cancer Development and ICI Efficacy

Changes in the gut microbiota have been associated with the development of cancer, as well as with immunotherapy response. First of all, dysbiosis could alter the gut mucosa, lead to chronic inflammation, and eventually promote colorectal cancer. Tabowei et al. noted in their review the predominant association of *Fusobacterium* genus with colorectal cancer and its depletion in healthy individuals [18]. Dysbiosis-associated microbes are thought to activate various immune pathways, including nuclear factor kappa B (NF-κB) dependent signaling, release of inflammatory mediators such as interleukins, or interaction with monocytes, promoting carcinogenesis in the mucosa epithelium [18]. Likewise, oral microbiota alterations leading to chronic periodontal inflammation have been associated with permanent genetic alterations in epithelial cells and eventually carcinoma development. Some of the suspected germs have been associated with inflammatory cytokines or TNF-α production, apoptosis inhibition, or promotion of cell proliferation. These oral pathogens have also been associated with colorectal and pancreatic cancer [19].

On the other hand, the influence of gut microbiota on cancer immunotherapy likely lies in its role in fine-tuning the immune status and in its possible activation against tumor cells under the action of ICIs [5]. It is thought to reduce Treg levels and promote CD8+ T cell activation and CD4+ T cell differentiation [20]. Two major studies in mice gave evidence of this strong interaction. First, the efficacy of anti-CTLA-4 was suppressed in antibiotic-treated and germ-free mice before being recovered through fecal transplantation from anti-CTLA-4-responding patients. It was discovered that these patients’ stools contained larger amounts of specific *Bacteroides* species. The same results were obtained through feeding the mice directly with *Bacteroides* isolates [21]. In addition, distinct preexisting microbiota have been associated with differences in spontaneous melanoma growth, which disappeared after fecal transplantation or cohousing the mice. Sequencing identified *Bifidobacterium* as a pivotal bacterial genus in cancer control, and oral supplementation with *Bifidobacterium* enhanced the efficacy of anti-PD-1 treatment [22]. In both cases, ICIs activity improvement is probably due to microbiome-induced maturation of dendritic cells and T cells’ priming and accumulation [21,22].

Other preclinical studies showed the benefit of probiotics on ICI efficiency in mice, notably through T cell induction [23]. Hibberd et al. reported probiotics-induced changes in gut microbiota composition in patients with colorectal cancer. These changes included higher levels of butyrate-producing bacteria, which are associated with a better response to immunotherapy [20,24]. Furthermore, a Japanese retrospective study suggests that probiotic supplementation by *Clostridium butyricum* could have a positive impact on progression-free survival and overall survival in patients with lung cancer receiving ICIs [25]. Finally, prospective trials are also currently recruiting to assess the benefit of fecal transplantation on response rates to ICIs; for example, the TACITO randomized controlled trial in patients with renal cell cancer (NCT04758507).

### 3.4. Focus on Akkermansia Muciniphila 

Routy et al. found significant differences in baseline fecal microbiota sequencing between patients with non-small cell lung carcinoma (NSCLC) or renal cell carcinoma (RCC) exhibiting long progression-free survival (PFS) under ICI (more than 6 months) or not. While further analyzing the stools’ characteristics of responders versus non-responders, they showed the overrepresentation of distinct bacterial taxa such as *Akkermansia* and isolated the species that was most significantly associated with a good response, *Akkermansia muciniphila* (*p* = 0.004). This anaerobe is one of the most abundant bacteria in the ileum microbiota. Its presence was identified in 69% and 58% of patients presenting partial response and stable disease, respectively, and 34% of those who progressed or died. The higher prevalence of this bacterium was then confirmed in a validation cohort. Responders’ stools conveyed sensitivity to PD-1 blockade after fecal transplantation to germ-free mice when non-responders’ did not [26].

The same group published a subsequent study to prospectively confirm the predictive nature of stool *A. muciniphila* on the response to ICIs (Derosa et al. [27]). In a multicentric cohort of 338 patients with NSCLC, baseline fecal *A. muciniphila* was associated with a better overall response rate (ORR: 28% vs. 18%, *p* = 0.04) and longer overall survival (OS) (18.8 months vs. 15.4 months, *p* = 0.03), independently of PD-L1 tumor proportion score (TPS), Eastern Cooperative Oncology Group Performance Status (ECOG-PS), or antibiotics therapy. The results were even more striking among the 86 patients receiving an ICI alone in first line with 41% ORR vs. 19% (*p* = 0.03) and 59% of long survivors (over 12 months) vs. 35% (*p* = 0.04). Interestingly, further stratification of patients according to their *A. muciniphila* levels (absent, low, high) revealed an association between previous antibiotic exposure and *A. muciniphila* overabundance, both correlated with decreased survival, suggesting that the importance of *A. muciniphila* needs a preserved microbial equilibrium. Finally, the team set up a prospective interventional phase 1/2 trial at Gustave Roussy Cancer Hospital to evaluate the contribution of *A. muciniphila* oral supplementation on the efficacy of ICI treatment for advanced NSCLC and RCC (CSET 3502, EV-2101, no NCT ID). Modulating the composition of the microbiota seems indeed to be a means of optimizing immune checkpoint blockade treatments [25,28].

The importance of *A. muciniphila* as a key gut bacterium has been evidenced in mice and humans aside from cancer therapy. Its lower abundance has also been associated with numerous diseases such as obesity, diabetes, or liver steatosis [29]. The favorable action of what is now considered as a “next-generation beneficial microbe” on metabolic diseases interestingly concerns the live bacterium as well as pasteurized *A. muciniphila* or some of its constituent parts (e.g., membrane protein or extracellular vesicles) [30]. However, the complex implications of this key bacterium and its derived products on the host’s metabolism and its perturbations is beginning to be understood and will be the focus of many further studies.

### 3.5. Role of Other Organ-Specific Microbiomes

In addition to the major role of the gut flora, studies have investigated the impact of other microbiota on the effectiveness of ICIs. Two organs in particular, the lungs and the bladder (until recently presumed sterile when healthy), could host a flora with a decisive role on the efficacy of ICIs. 

New generation sequencing (NGS) or 16S ribosomal RNA sequencing of various sputum samples, bronchoalveolar lavage, or lung tissue have led to the detection of many bacterial species belonging to specific taxa such as *Prevotella*, *Streptococcus,* or *Veillonella*. Even though around 90% of identified DNA likely comes from nonviable or dead microorganisms, some studies discovered culturable bacteria in lung samplings [31]. Changes in this microbiota in childhood have been associated with several lung diseases like asthma, and other changes can occur under inhaled corticosteroid treatments [32]. This may play an important role as well in carcinogenesis, and it has been suggested that patients with lung cancer present decreased diversity and higher rates of dysbiosis-associated bacterial genera like *Granulicatella* or *Abiotrophia*. Dysbiosis also seems to be associated with decreased immune modulation, especially through γδ T cells, and subsequently with loss of cancer control. Although it seems conceivable that the impact of antibiotics on the loss of efficacy of ICIs may also involve the lung flora in addition to the gut microbiota, no specific studies have been conducted to date [31].

The same sequencing techniques applied to healthy individuals’ culture-negative urine samples isolated numerous bacteria—noticeable taxa were *Lactobacillus*, *Corynebacterium,* or *Streptococcus*—overturning the older dogma of Louis Pasteur and affirming the existence of the urinary microbiota. Similarities in urinary bacterial flora have been reported in patients with prostate cancer, which differed from healthy controls—without differences of gut microbiota. This contributes to the hypothesis of an etiopathogenic role of the urinary dysbiosis in carcinogenesis, the involvement of chronic infection and inflammation being already well-known with schistosomiasis. It has also been suggested that as well as the inflammation caused by tobacco or other carcinogenic agents, urinary dysbiosis could be associated with high tumor mutational burden, and could therefore increase susceptibility to immunotherapy. However, the real impact of this newly identified microbiota and its actions on the efficacy of ICIs has not been formally studied yet [33]. 

The potential role of oral microbiota on head and neck squamous cell carcinoma (HNSCC) incidence and management has also been studied [34]. Hayes et al. showed an association between greater oral abundance of commensal *Corynebacterium* and *Kingella* spp. and a decreased risk of HNSCC occurrence in a nested case-control study [35]. However, regarding ICI efficacy, Ferris et al. found no significant impact of oral microbiota diversity on nivolumab (anti-PD-1) response in 85 patients with HNSCC from the CheckMate 141 trial [36].

### 3.6. Treatment-Induced Microbiome Changes

Unlike the genome inherited from the parents, the microbiome involves a complex ecosystem that is unstable and can be significantly altered throughout life, especially by age, diet, disease, or drugs. The interaction between gut flora and medications is intricate and bidirectional: drugs can alter the microbiome composition, but gut microorganisms also influence the host response to the drug through what is referred to as pharmacomicrobiomics [20].

One major recent discovery involves the impact of many non-antibiotic drugs on microbiota composition, and thus on the health condition of the host, notably its immune status. Some of these drugs are widely used, such as proton pump inhibitors, metformin, or statins, while others are used more specifically in oncology, such as corticosteroids, opioids, or laxatives [5]. Major studies have reported drug-induced changes in microbiome constitution using large population-based cohorts in the Netherlands [37], the UK [38], or Belgium [39]. Up to 19 commonly used drugs, including additional medications to those mentioned before, from other families such as beta blockers, aspirin, or angiotensin-converting enzyme inhibitors, have been shown to impact microbiome diversity, increasing or decreasing specific bacterial species [5]. Because many patients are taking multiple medications, some studies have examined the additional effect of polypharmacy [40,41].

## 4. Corticosteroids

Due to their strong immune modulating properties, including cytokine release alteration and T cell activation, migration, and differentiation inhibition [42], corticosteroids are widely used for the treatment of inflammatory and autoimmune diseases. However, they are also broadly used in oncology, prescribed from short to long courses of treatment, in small or large doses, to treat many cancer-related symptoms: pain, nausea, dyspnea, occlusion, cerebral or meningeal symptoms, etc. Related adverse events of such medications are well known and entail infections, including opportunistic infections like pneumocystis, osteoporosis, diabetes, glaucoma, cataracts, or skin alteration [43]. 

### 4.1. Impact of Immunosuppression on Cancer Development and Microbiota Composition

Complex interactions between the immune system and cancer development have been studied through observations in patients with primary or acquired immunodeficiencies (e.g., organ transplant). The concept of cancer immunosurveillance involves, notably, interferon production and T cells promotion. Newly produced cancer cells are physiologically identified and killed by natural killer cells and cytotoxic T cells, and if tumor cell proliferation escapes immune monitoring, cancer growth remains immunologically restricted. However, both phases can be altered in immunosuppressed patients [44]. Chronic use of glucocorticoids has been associated with the occurrence of cancers such as lymphomas or sarcomas, especially Kaposi’s sarcoma in transplant and non-transplant clinical settings [45,46].

Immunosuppression-induced dysbiosis has been mostly studied in solid organ transplant recipients with limited outcomes and biases involving the complex interactions between pre-transplant diseases (e.g., cirrhosis) and the gut microbiota. The main results were reduced *Enterobacteriaceae* and increased *Lachnospiraceae* and *Ruminococcaceae* after liver transplant and higher abundance of Bacteroidetes in patients without diarrhea after kidney transplant [47]. Moreover, immunosuppressive drugs, including prednisone, have also been associated with decreased Bacteroidetes and increased Firmicutes in mice [48]. Both acute and chronic dexamethasone treatments have been associated in mice and rats with shifts in gut microbiota [49] and increased levels of *Bifidobacterium* and *Lactobacillus*, and it is remarkable to note that the anti-inflammatory action is passed on after fecal transplantation from a treated individual to one with a genetic susceptibility for IBD [50]. Likewise, the use of inhaled or systemic corticosteroids for asthma, chronic obstructive pulmonary disease (COPD), or chronic rhinitis has been associated with changes in diversity, composition, and/or burden of the respiratory microbiome [32]. However, the impact of glucocorticoids on ICI treatment response involves, more likely, direct immune modulation, rather than microbiome-induced [42].

### 4.2. Effect on ICI Treatment Efficacy

Patients receiving systemic glucocorticoids over 10 mg of prednisone equivalent were usually excluded from ICIs clinical trials because this dose is considered immunosuppressive. However, numerous real-world studies analyzed the impact of such medication at baseline or during ICI treatment and showed a significant decrease in PFS and OS in patients on glucocorticoids [51,52]. Ricciuti et al. conducted a retrospective study to measure the indication-related impact of steroid therapy, which appears to be more detrimental on survival if initiated for palliative care (e.g., fatigue, pain, brain edema) than to treat non-cancer-related diseases (e.g., COPD flare, auto-immune disease), even at high doses [53]. Table 1 and Table 2 present the results of studies (mostly retrospective, except for Hendriks et al.) and systematic reviews and meta-analyses, respectively, on this subject, providing subgroup analyses on the indications of steroid treatment. The same differences were observed between palliative care indications or brain metastases subgroups, and between non-cancer-related indications or immune-related adverse events (irAEs) management. The deleterious impact on survival could therefore be linked to the preexisting poor prognosis of the patients needing steroids for palliative reasons or brain metastases rather than to the action of steroids themselves [42]. However, even among patients with baseline brain metastases, the observed deleterious effect of corticosteroids could reflect the pejorative nature of symptomatic cerebral lesions. On the contrary, association of irAEs with better overall prognosis has often been reported and could compensate for possible harmful effects of glucocorticoid prescription. 

These findings suggest reassuring data for clinicians about the possibility of initiating corticosteroids even at high dosage for irAE management, but could lead to delay immunotherapy or reconsider other treatment options like chemotherapy for patients requiring high dose palliative steroids if no withdrawal is possible before initiation.

## 5. Antibiotics

Early on, antibiotics were suspected in dysbiosis-induced variations of immune response because of their strong impact on the composition, diversity, and burden of gut microbiota. Due to their long-term influence on metabolic pathways, even after early-age short treatments, they have been associated with various chronic diseases in mice and in humans, such as asthma or obesity [62,63]. The increasing evidence of the potentially detrimental effect of antibiotics on immunotherapy results has led to the exclusion of patients who received recent antibiotic therapy from clinical trials or to the requirement of antibiotic wash-out periods.

### 5.1. Antibiotic-Induced Perturbations of the Microbiota

Among the many classes of antibiotics, not all have the same impact on the intestinal flora. The differential action on the families of bacteria will disturb the balance and diversity of this complex microenvironment for a long time, and in doing so, uncouple the mutualistic host-microbiota relationship. While nearly full recovery is often obtained after four weeks, several studies have shown that the consequences of an antibiotic treatment can be observed in the long term on the composition of the microbiota. Most durable perturbations occur with antibiotics having broad activity on anaerobes, like clindamycin [64]. Jakobsson et al. explored throat and fecal microbiota after clarithromycin and metronidazole treatment. While the flora of control individuals showed relative long-term stability, treated patients showed profound changes in the microbiota within a week of treatment, and alterations were still noticeable up to four years later. These changes could affect the bacterial composition (e.g., Actinobacteria decrease), but also the latent presence of resistance genes selected earlier (e.g., *erm*(B) coding for macrolide resistance) [65]. Besides the flora’s composition and selection of potential resistances, the complex alterations involve interactions between bacterial species themselves and favorable biochemical conditions for pathogenic species’ proliferation. Moreover, if antibiotics induce broadly human-conserved microbiota alterations, some of their consequences include unpredictable host-specific responses leading to so-called “ecological surprises” [64].

Antibiotics have also been associated with the development of colorectal cancer (CRC). Simin et al. conducted a large meta-analysis including more than 4 million individuals and over 73,000 CRC. Higher risk of CRC was associated with antibiotics, particularly for broad-spectrum antibiotics. The causality is not clearly established, and could involve DNA-damage (e.g., quinolones), treatment-induced dysbiosis, or pathogenic colonization promoting the production of DNA-damaging toxins [24,66]. These metabolites include notably bacterial-derived genotoxins (e.g., BFT from *Bacteroides fragilis* or colibactin from *Enterobacteriaceae*) and bacterial virulence factors (e.g., FadA from *Fusobacterium nucleatum*) that have been found to activate cancer-promoting signaling pathways or cause DNA damages [67].

### 5.2. Impact of Antibiotics on ICI Response according to Histology

Many retrospective studies have analyzed the effect of antibiotics on response rate and survival among patients receiving ICIs, and the results of some of them are compiled in Table 3. Antibiotics seem to alter progression-free and overall survival regardless of the histology of cancer. Routy et al. found decreased OS in patients with NSCLC, confirmed in NSCLC validation cohort, and decreased PFS in patients with renal cell cancer and urothelial cancer [26]. These results were confirmed in the same team’s prospective NSCLC cohort: exposure to antibiotics was associated with poorer survival in both groups exhibiting fecal *Akkermansia muciniphila* or not [27]. 

### 5.3. Specificities under Immunotherapy versus Chemotherapy

Cortellini et al. added a second cohort in their retrospective study of patients treated with chemotherapy alone. Interestingly, the use of antibiotics has no impact on chemotherapy efficacy [71]. Although patients in the immunotherapy arm were significantly older and more frequently with an ECOG-PS ≥ 2, these results suggest a real impact of antibiotics on specific immune response rather than a bias towards survival due to initially poorer prognosis of patients needing such treatments. Cortellini et al. published another international retrospective study to estimate the effect of antibiotics on response and survival in patients with NSCLC treated with a first-line chemo-immunotherapy combination [72]. No association was found with this treatment modality, even among patients with a high TPS score of 50% or more. Among the antibiotic therapies preceding antitumor treatment, the modality of administration (intravenous or oral) or duration (≥ or <7 days) did not show any significant impact either. 

### 5.4. Importance of the Antibiotic Treatment Modality: Timing and Duration

Table 4 summarizes some systematic reviews and meta-analyses on the subject, including subgroup analyses on histology and/or chronology of antibiotics and ICI treatment. If no major differences seem to emerge from the type of cancer, the numerous studies highlight one discernible point on the timing of antibiotic therapy in relation with the initiation of treatment with ICI. There is apparently a window of maximum deleterious effect, approximately between the month before and the month after the beginning of immunotherapy. The importance of this pivotal period probably reflects the lasting impact of dysbiosis-induced immune “priming” perturbation [73].

Moreover, Tinsley et al. included subgroup analyses in their retrospective study on single versus cumulative antibiotic therapies [70]. Results show a significant decrease in survival only with cumulative treatments defined as concurrent or successive antibiotics for more than seven days. Once again, it is likely that the patients in need for these treatments were in a more severe condition, which could explain the results, regardless of the possible impact on the response to ICI. 

### 5.5. Importance of the Antibiotic Treatment Modality: Molecule and Spectrum

Antibiotics are a very large family of molecules, presenting different pharmacokinetics, pharmacodynamics, and activity spectra. In vitro studies showed maximal impact on microbiota with antibiotics having a strong activity on anaerobes (e.g., macrolides). Some studies have been able to differentiate the action of antibiotics on the results of immunotherapy according to their class. Ahmed et al. showed no impact on response rate with narrow Gram-positive spectrum antibiotics (vancomycin, daptomycin, linezolid), whereas broad spectrum antibiotics (e.g., β–lactams, quinolones, cyclines) affected both response rate and PFS [78]. Chalabi et al. examined the difference in OS between chemotherapy and atezolizumab (anti-PD-L1) for several antibiotic classes without showing any significantly impairing class for ICIs survival [79]. Because amoxicillin (AMX) is a widely used penicillin, alone or in association with clavulanic acid (AMC), it is interesting to note that Gaucher et al. found no difference in OS between patients treated with AMX versus AMC, nor between those treated with AMX or AMC versus other antibiotics [6].

However, Medik et al. presented interesting outcomes in mice treated with anti-CTLA-4 ICI and metronidazole. The antibiotic-induced shift in microbiome composition was associated with a favorable immune microenvironment and more complete tumor regression [80]. Using antibiotics to modulate the immune response through bacterial changes could represent an attractive way to improve the action of ICIs [81]. This has been explored with oral vancomycin in liver tumors: its action on bile acid metabolism has been associated with chemokine ligand CXCL16 expression. Monge et al. are conducting a Phase 2 single-arm trial of nivolumab with oral vancomycin and tadalafil, a phosphodiesterase inhibitor, in hepatocellular carcinoma (HCC) and liver-dominant metastatic digestive cancers (NCT03785210) [82].

## 6. Proton Pump Inhibitors

Proton pump inhibitors (PPIs) are very commonly used to treat peptic ulcers or gastroesophageal reflux disease or to prevent digestive complications due to non-steroidal anti-inflammatory drugs. Because of their low level of toxicities, they are often over-prescribed: up to 70% of PPIs prescriptions may be unnecessary and not grounded on evidence-based recommendations. Moreover, initial indication is rarely re-evaluated, leading to unjustified chronic use [5]. Since their arrival on the market in 1989, they have been suspected in association with numerous adverse events such as hypomagnesemia, vitamin B12 deficiency, bone fracture, or *Clostridioides difficile* infection [83]. Because oncology patients in particular often present previous comorbidities, inherent frailty, and potentially ulcer-inducing comedications, they are quite frequently treated with PPIs. 

### 6.1. PPI-Induced Alterations of the Microbiota

Increased gastric pH caused by acid secretion blockade can lead to downstream disturbances in the equilibrium of gut flora, but PPIs can also generate pH-unrelated effects. Indeed, they are associated with hormonal disturbances (e.g., hypergastrinemia, hyperparathyroidism) and alterations in nutrient absorption [84]. Several studies and large population-based cohorts led to evidence of profound alteration in microbiota composition and diversity under PPI treatment [5,37]. These perturbations involve decreased gut commensal bacteria (e.g., Ruminococcacae, Bifidobacteriaceae) to the benefit of oral cavity microorganisms (e.g., *Rothia dentocariosa*, *Rothia mucilaginosa*, *Actinomyces* spp.) [5,85]. 

PPIs also induce favorable conditions for colonization by enteric pathogens leading in PPI-users-to-odds-ratio of 1.5 to 1.8 for *C. difficile* and 2 to 4 for other germs such as *Campylobacter* or *Salmonella* [86]. This pathogen spread might be promoted by alterations in colonic colonization resistance patterns, such as increased *Lactobacillus* spp. and decreased Bacteroidetes [84]. Similar alterations in bacterial orders have been found in PPI use-discordant monozygotic twins from the TwinsUK cohort, and with small interventional study [85]. Finally, PPIs are also associated with a 3-fold increase in risk of small intestinal bacterial overgrowth (SIBO), involving species like *Escherichia coli*, *Enterococcus* spp., or *Klebsiella pneumoniae* [84].

If gut flora perturbations contribute to the negative impact of PPIs on ICI survival, retuning the microbiota could be a means to restore ICI efficacy. Indeed, Tomita et al. reported significantly better OS and PFS in a small retrospective study among PPI-using patients with NSCLC treated with ICIs when receiving oral supplementation with *Clostridium butyricum* (OS: HR 0.42 [95% CI: 0.19, 0.92] *p* = 0.03; PFS: HR 0.52 [0.29, 0.94] *p* = 0.03). The oral supplementation was also associated in patients’ stools with decreased oral pathogens (e.g., *Atopobium*, *Streptococcus*) as well as increased gut bacteria *Bifidobacterium* spp., which has been associated with ICI efficacy in previous studies [87].

### 6.2. Impact on ICI Efficacy

Apart from their activity on the gastrointestinal microbiome, PPIs are thought to directly alter the inflammatory response, especially by decreasing the secretion of adhesion molecules by inflammatory cells and by inhibiting cytokines production [73]. Results of studies of PPIs’ impact on response and survival under ICI treatment are heterogeneous, as illustrated in Table 5. First studies looking for a detrimental impact of PPIs on ICI efficacy were post hoc analyses from pooled phase 2 and 3 randomized controlled trials. Pooled patients in atezolizumab arms presented significantly decreased OS and PFS when receiving prior or concomitant PPI, whereas survival in chemotherapy arms was not altered [79,88]. Subsequent large-scale retrospective studies are also discordant. Stokes et al. found no impact of PPIs on survival in more than 3500 ICI-treated veterans with NSCLC [89], while Baek et al. studied nearly 3000 patients with NSCLC receiving ICIs from the South Korean national health system and evidenced a significant decrease in OS due to PPI use [90]. Moreover, they evidenced stronger pejorative effect in “new PPI users”, defined as patients having initiated PPIs after a washout period of at least 180 days, suggesting a more deleterious action when the first PPI intake in a naive organism is closer to the initiation of ICI.

### 6.3. Differences in Histology

Large meta-analyses have also led to divergent results as shown in Table 6. Subgroup analyses yielded some surprising results, like an inconsistently favorable effect of PPI use on survival in patients with melanoma. The two meta-analyses use the same retrospective studies, including the results of Failing et al. on 159 patients with melanoma at Mayo Clinic, whose results found a trend in this direction, although not significant [92]. All the meta-analyses seem to agree on decreased survival in patients with NSCLC [93,94,95,96], whereas no significant impact has been shown in patients with RCC, based on a pool of more than 400 patients [95]. 

### 6.4. Importance of Timing

As well as with antibiotics, many studies reported in Table 1 evidenced a critical window associated with more negative results. This period involves about 30 days before and after the start of immunotherapy. Subgroup analyses investigating the effect of PPIs after initiation of ICI treatment do not evidence significant decreased survival [94,95]. Thus, the importance of a preserved gut microbiota during initial immune priming can once more be hypothesized.

## 7. Other Medications according to Pathway Alterations

### 7.1. Metabolism and Hypoxia Lowering: Metformin

Metformin is a widely prescribed blood glucose-lowering treatment and standard of care for type 2 diabetes. It amplifies insulin sensitivity and inhibits liver gluconeogenesis. Its complex mechanism of action is not entirely understood and is thought to involve the gut microbiota: interventional studies showed gut flora alteration in healthy volunteers receiving metformin. Interestingly, fecal transplantation from metformin-treated individuals induced lower blood glucose levels in germ-free mice [5,97]. Metformin especially favors the abundance of *Akkermansia muciniphila*, which independently enhances glucose tolerance in mice [98]. 

Its action on the mitochondrial metabolism chain also impacts the immune system by decreasing immune exhaustion and stimulating IL-10 secretion and CD8+ T-cells promotion, as well as cancer cells by diminishing intratumoral hypoxia [73,97,99]. Metformin also interacts with the PD-1/PD-L1 checkpoint pair: the degradation of membrane PD-L1 after metformin treatment has been evidenced in breast tumor mice models and confirmed in breast cancer samples of metformin-treated patients [100]. In vitro and animal experiments showed increased CD8+ T cell infiltration and enhanced efficacy of PD-1 inhibitor pembrolizumab in lung cancer when associated with metformin [101]. Moreover, intratumoral hypoxia has been associated with decreased efficacy of PD-1 blockade in murine models. That barrier to immunotherapy could be reduced by metformin-induced diminution of oxygen consumption in tumor cells, allowing for higher infiltration by T cells and better tumor response [102]. 

However, the objective impact of metformin on ICI efficacy in clinical practice has not been formally proven thus far. Afzal et al. evidenced longer OS and PFS in a small retrospective cohort of ICI-treated patients with metastatic melanoma receiving concomitant metformin (22/55 patients) without reaching statistical significance (OS: 46.7 m vs. 28 m, HR 0.40 [95% CI: 0.12, 1.35] *p* = 0.12; PFS: 19.8 m vs. 5 m, HR 0.55 [0.24, 1.24] *p* = 0.15) [103]. The same team published similar trends with patients with NSCLC: non-significant improvement in OS and PFS was found with concomitant metformin (21/50 patients; OS: 11.5 m vs. 7.6 m, HR 0.8 [95% CI: 0.39, 1.63] *p* = 0.5; PFS: 3 m vs. 4 m, HR 0.86 [0.47, 1.6] *p* = 0.6) [104]. Recently, Yang et al. found analogous results in a bigger cohort of 466 NSCLC patients. Eighty-nine patients (19%) received metformin from at least 8 weeks before ICI initiation. The study showed significantly higher ORR (24.7% vs. 14.8%, *p* = 0.025) and PFS (5.1 m vs. 2.8 m, HR 0.69 [95% CI: 0.52, 0.93] *p* = 0.013) without a difference in OS [105].

Phase 1 and 2 trials are currently open to prospectively assess a positive effect of concurrent metformin on anti-PD-1 and anti-PD-L1 treatment for different types of cancer: melanoma (NCT04114136), NSCLC (NCT03048500), or microsatellite stable (MSS) colorectal cancer (NCT03800602) [97,106,107].

### 7.2. Local Inflammation: Aspirin and Nonsteroidal Anti-Inflammatory Drugs

Aspirin and nonsteroidal anti-inflammatory drugs (NSAIDs) act as more or less selective inhibitors of cyclooxygenases (COX1/COX2). These enzymes are involved in prostaglandins synthesis, including prostaglandin E2, which has been associated with tumor cell survival, growth, and invasion [108,109]. Moreover, in vivo studies showed the role of COX2 in ICIs resistance, notably by decreasing the infiltration of immune cells in the tumor environment. Pi et al. showed experimental reverse of pembrolizumab resistance in mice when combined with aspirin or NSAID celecoxib [109]. The benefit of aspirin on the response to ICIs could also involve its antiplatelet action: Riesenberg et al. showed improved anti-PD-1 response in vivo with aspirin and another antiplatelet agent, clopidogrel: tempering platelet activation could generate beneficial immune remodeling in the tumor microenvironment [110].

Once again, the transposition of these promising findings to patients has not yet yielded satisfactory results. Different studies specifically examined the impact of aspirin and/or NSAIDs on response and survival on ICIs [111,112,113]. The results, although not statistically significant, all support a deleterious association. However, the prescription of NSAIDs in oncology patients may be a significant bias due to their use for cancer pain and could reveal more active, aggressive, or rapidly progressing disease that would necessarily impair survival. In some studies, looking for several drugs’ impact on ICI survival that isolated low-dose aspirin from NSAIDs prescription, a favorable effect has been evidenced with aspirin. Cortellini et al. showed higher ORR (OR 1.47 [95% CI: 1.04, 2.08] *p* = 0.03) and longer PFS (HR 0.79 [0.64, 0.98] *p* = 0.03) in a 1012-patient retrospective study [114]. Zhang et al. compiled five studies with specific low dose aspirin subgroups in their meta-analysis and evidenced longer PFS (HR 0.84 [0.72, 0.98] *p* = 0.02) with no impact on OS [115].

### 7.3. Stress and Neuro-Oncology: Beta Blockers

The role of the β-adrenergic receptor has been recently highlighted in the tumor microenvironment, as well as its potential influence on cancer growth, invasion, and metastatic spreading [116,117]. The β-adrenergic receptor blockers, or beta blockers, are widely used treatments for hypertension, coronary heart disease, or arrhythmia. Because of the negative impact of stress-induced adrenergic signaling on the immune system, beta blockers have been thought to positively impact ICI efficacy. They have been reported to decrease tumor neoangiogenesis and to improve anti-CTLA-4 activity in mice models [118]. Kokolus et al. showed significantly higher OS in patients with metastatic melanoma receiving ICIs and/or interleukin 2 with concomitant beta blockers, and confirmed the results in a preclinical murine model [119]. Mellgard et al. found better disease control (ORR: OR 2.79 [1.54, 5.03] *p* = 0.001) with no impact on survival in a 339-patient cohort [120]. Similar results have been shown in other retrospective studies and meta-analyses [114,117]. Oh et al. evidenced beta blocker-associated longer PFS (HR 0.58 [0.36, 0.93]) without benefit on OS in a small retrospective study of 109 patients with NSCLC [121].

Finally, a phase 1 study by Gandhi et al. established a treatment dosing of 30 mg propranolol twice a day combined with pembrolizumab with satisfactory safety in melanoma [122]. Phase 2 trials are currently recruiting in order to test this potential positive association in triple negative breast cancer (NCT05741164), urothelial carcinoma (NCT04848519), or melanoma (NCT03384836). Again, biases on tumor response and survival are possible with beta blockers, even if they appear less obvious than with other molecules.

### 7.4. Microenvironment Remodeling and Immune Modulation

#### 7.4.1. Renin-Angiotensin-Aldosterone System Inhibitors

The renin-angiotensin-aldosterone system (RAAS) is a key target for hypertension treatment. It is also thought to enable cancer cell growth and proliferation through its interaction with the tumor microenvironment, especially cancer-associated fibroblasts (CAFs). Angiotensin-receptor blockers (ARB) have been shown to reprogram CAFs to an immunosupportive state and to diminish T cell inhibition and immunosuppressive factor expression (e.g., interleukins 6 and 10) [123,124,125]. Moreover, RAAS inhibitors’ action on the tumor microenvironment involves blood vessel decompression and therefore better tumor perfusion, allowing for hypoxia alleviation and better drug delivery [126]. Chauhan et al. designed a tumor-selective ARB that has been associated with improved tumor T cell activity and increased ICI efficacy in animal models without blood pressure-lowering effects [125]. 

The results of real-life studies are divergent. Nuzzo et al. showed encouraging results in metastatic RCC with increased OS (HR 0.35 [0.17, 0.70] *p* = 0.003) and longer time to treatment failure (HR 0.57 [0.36, 0.92] *p* = 0.02) [127]. Drobni et al. showed higher OS with concurrent RAAS inhibitors in a large retrospective cohort of almost 6000 ICI-treated patients receiving diverse anti-hypertensive medication (HR 0.92 [0.85, 0.99] *p* = 0.03) [128]. However, Medjebar et al. found an opposite effect in a 178-patient retrospective study, with a decrease in PFS in patients receiving angiotensin-converting enzyme (ACE) inhibitors. Subsequent RNA sequencing of tumor samples suggested an immunosuppressed state in the ACE inhibitors group with a lower rate of M1 macrophages, activated mast cells, NK cells, and memory-activated T cells [129]. Further studies—ideally prospective—are still needed to answer the question and to prove a benefit in clinical practice.

#### 7.4.2. Opioids

Strong analgesia is often needed due to intense cancer pain: opioids are therefore widely used in oncology. Morphine-like analgesics include numerous molecules (e.g., morphine, fentanyl, oxycodone, codeine) that have been thought to stimulate tumor progression and metastasis formation because of overexpressed opioid receptors on cancer cells. Their role has also been shown in impairing the immune system through T cell modulation and in gut flora alterations [130]. Hence, the impact of opioid treatments on ICI efficacy was the focus of several retrospective studies [131,132,133] and meta-analyses [134]. The results seem to be in favor of lower response rates and shorter PFS and OS. 

Yet the breadth of difference in response (e.g., in Tanigushi et al. 2.6% vs. 13.5%) [132] or survival (e.g., in Botticelli et al. 3 vs. 19 months of PFS, and 4 vs. 35 months of OS) [133] suggests the existence of strong biases. The necessity of opioid use is indeed likely associated with poorer general condition, aggressive disease, or pejorative metastatic burden, and studies with a higher level of evidence, if possible, will be needed to conclude a real effect.

#### 7.4.3. Statins

Statins are cholesterol-lowering agents commonly used in the primary and secondary prevention of cardiovascular disease. They have been thought to potentially interfere with ICI efficacy due to their immune modulation properties. In preclinical models, statins have been shown to inhibit PD-L1 expression on cancer cells by action on multiple signaling pathways such as RAS, AKT, or ꞵ-catenin. Their activity also involves enhanced T cell activity in tumor microenvironments as well as in draining lymph nodes [135,136,137]. Retrospective studies found improved survival in malignant pleural mesothelioma and NSCLC [138,139], but meta-analyses are discordant. Yongchao Zhang et al. showed longer OS and PFS based on five studies (OS: HR 0.76 [0.63, 0.92] *p* = 0.005; PFS: HR 0.86 [0.75, 0.99] *p* = 0.036) [115], while Lei Zhang et al. found no significant association between statin use and OS or PFS based on eight cohorts including 2382 patients [140]. Again, the evidence for a real impact on the efficacy of immunotherapy treatment is limited, and further study is needed.

## 8. Discussion

We showed that many studies have explored the potential impact of concurrent use of multiple therapeutics on the outcomes of ICI treatment, from in vitro and animal models to retrospective clinical studies. Described interactions are very diverse and can positively or negatively affect various cancer-promoting mechanisms, as well as several necessary factors for the functioning of ICIs. Figure 2 illustrates these complex linkages and interactions with some of the hallmarks of cancer proposed by Hanahan and Weinberg [141].

Positive or negative associations have been evidenced regarding response and survival outcomes, and it seems difficult for the clinician to determine the right therapeutic attitude in the patient’s best interest. Moreover, direct extrapolation of convincing in vitro or animal models’ results is not possible and proper randomized controlled trials cannot always be conducted for such issues. It is therefore necessary to rely on well-conducted retrospective studies and meta-analyses. They are not immune to many biases, in particular on the probability of a more precarious general condition due to comorbidities in patients requiring comedications, which independently affects their prognosis.

The results we gathered seem to show a detrimental association of corticosteroids given for supportive care reasons, as well as antibiotics and PPIs on objective response and survival, especially when used within a window of a few weeks before and/or after the start of ICI therapy. The first results concerning the other comedications (metformin, NSAIDs, beta blockers, RAAS antagonists, opioids, and statins) also call for caution. It should be kept in mind that any new drug introduction may have an unpredictable effect on expected oncologic outcomes and should not be minimized.

Yet, it is not always possible for the clinician to stop the treatments involved because of their often unquestionable necessity. One should however try to manage the timing of treatment as well as possible, even sometimes by delaying the ICI by a few weeks to allow time for the microbiota to recover and to ensure that the immunotherapy has not been initiated in a particularly deleterious period.

One should also ascertain the need to introduce a new concomitant treatment, according to evidence-based recommendations: for instance, avoiding prophylactic antibiotics or unjustified PPIs. Countermeasures like phages or histamine H2-receptor antagonists are suggested by Derosa et al. among other “microbiota-centered interventions” in immuno-oncology (e.g., intermittent fasting, fiber intake, aerobic and anaerobic physical exercise) [28]. 

If deleterious comedications are required, the possibility to initiate alternative oncological treatments such as chemotherapy or targeted therapy should be discussed. Buti et al. developed and validated a prognostic index based on OS under ICI treatment in patients with baseline comedications which might be helpful. Corticosteroids are assigned 2 points, antibiotics and PPIs 1 point each, generating a 3-risk group stratification that significantly predicted OS, PFS, and ORR in an external validation cohort: score 0 (good prognosis), score 1–2 (intermediate), 3–4 (poor) [142].

Finally, looking to the future of immunotherapy for solid tumors leads to a consideration of new ICI targets (e.g., TIGIT, LAG-3, TIM-3), new combinations [143], and new treatment classes such as vaccines and chimeric antigen receptors T cells (CAR T cells). Questions will arise as to their efficacy in real-life conditions, and studies must be conducted in order to know whether the interactions observed with anti-PD-(L)1 and anti-CTLA-4 will be confirmed with these new strategies. 

## 9. Conclusions

We reported here the complex associations between the microbiota, the immune system, and immunotherapy response, as well as the promising results expected from microbiota adjustments, notably with the key commensal bacterium *Akkermansia muciniphila*. We gathered data on the negative impact of corticosteroids, antibiotics, and proton pump inhibitors on ICI efficacy, highlighting the critical period surrounding the initiation of immunotherapy and the interesting differences between palliative or non-palliative corticosteroids indications, narrow or broad spectrum antibiotics, and former or new PPIs users. We also gave the preliminary outcomes concerning the positive impact on ICI efficacy of several frequently prescribed medications such as metformin or beta blockers, although these conclusions need to be assessed in prospective trials. 

Our review provides a practical tool for the clinician, showing the critical involvement of the microbiota in the efficacy of ICIs and the impact of many usual comedications on the oncological response. These results are intended to help clinicians for overall management of patients receiving immunotherapy, taking into account all the patient’s comedications to support these complex decisions.

## Figures and Tables

**Figure 1 cancers-15-02276-f001:**
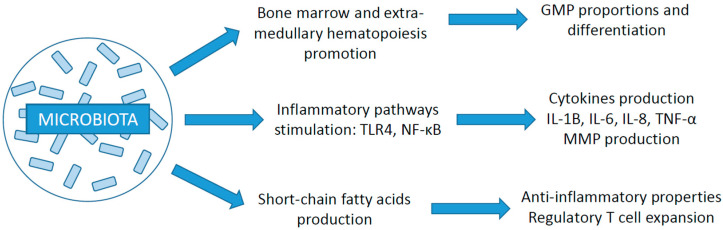
Schematic diagram summarizing the relation between the microbiota and the immune system (GMP: granulocyte and/or monocyte progenitor; TLR4: Toll-like receptor 4; NF-κB: nuclear factor kappa B; IL: interleukin; TNF: tumor necrosis factor; MMP: matrix metalloproteinases) [15,16,17].

**Figure 2 cancers-15-02276-f002:**
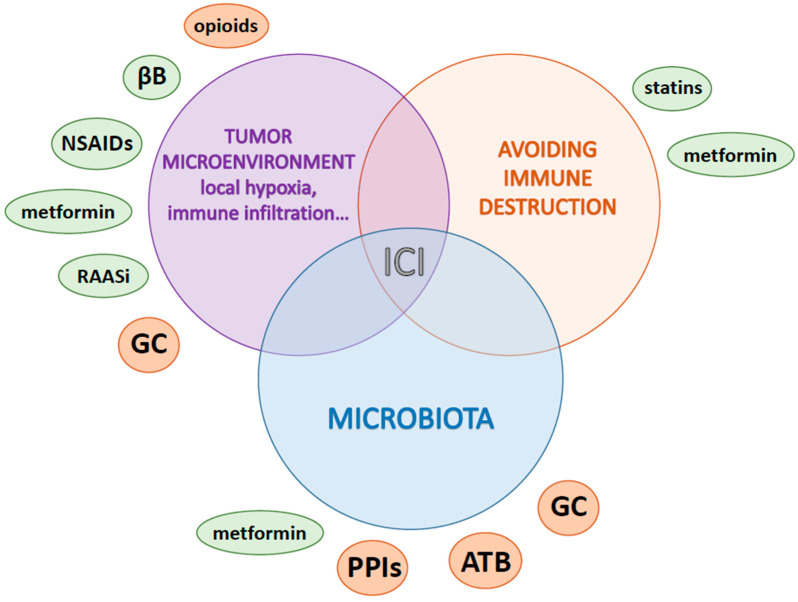
Multiple interactions between drugs, cancer hallmarks and ICI efficacy (red/green: negative/positive impact on ICI efficacy, respectively; ICI: immune checkpoint inhibitor; GC: glucocorticoids; ATB: antibiotics; PPIs: proton pump inhibitors; NSAIDs: non-steroidal anti-inflammatory drugs and aspirin; ꞵB: beta blockers; RAASi: renin-angiotensin-aldosterone system inhibitors).

**Table 1 cancers-15-02276-t001:** Studies analyzing the impact of baseline or concomitant corticosteroids with ICI treatment on response rate and survival [51,52,53,54,55,56,57,58,59].

Author (Year)	Type of Cancer	ICI	GC Regimen and Indication	*n* Patients/Total (%)	Compared Arms	ORR [CI 95%]	PFS [CI 95%]	OS [CI 95%]
Margolin (2012) [54]	Melanoma with BM	Ipi	Systemic GC for symptomatic BM	21/72 (29)	No statistical comparison	-	1.3 m vs. 2.7 m *	3.7 m vs. 7 m *
Chasset (2015) [55]	Melanoma	Ipi	≥10 mg pred baseline multiple indications	12/45 (27)	Overall	-	-	4 m vs. 11 m HR: 5.82 [2.45, 13.8], *p* < 0.001
					Among BM	-	-	4 m vs. 7 m, *p* = 0.043
Arbour (2018) [51]	NSCLC	Multiple	≥10 mg pred baseline multiple indications	90/640 (14)	Overall	7% vs. 18%, *p* = 0.05	HR: 1.31 [1.03, 1.67], *p* = 0.03	HR: 1.66 [1.28, 2.16], *p* < 0.001
Scott (2018) [52]	NSCLC	Nivo	(A) ≥10 mg pred in first 30 d	25/210 (12)	A vs. non A	-	-	HR: 2.30 [1.27, 4.16], *p* = 0.006
			(B) ≥10 mg pred for irAEs	31/210 (15)	B vs. non B	-	-	16.1 m vs. 10.5 m, *p* = 0.50
Hendriks (2019) [56]	NSCLC BM+ or BM− prospective	Multiple	(A) GC at baseline (overall)	141/1025 (14)	A vs. non A	-	HR: 1.31 [1.07, 1.62], *p* = 0.01	HR: 1.46 [1.16, 1.84], *p* = 0.001
			(B) GC at baseline among BM+	69/255 (27)	B vs. BM + non B	-	HR: 2.78 [1.90, 4.08], *p* < 0.001	HR: 2.37 [1.54, 3.63], *p* < 0.001
Ricciuti (2019) [53]	NSCLC	Multiple	(A) ≥10 mg pred baseline: SC	56/640 (10)	A + B vs. C	10.8% vs. 19.7%, *p* = 0.04	2.0 m vs. 3.4 m HR: 1.36 [1.08, 1.73], *p* = 0.01	4.9 m vs. 11.2 m HR: 1.68 [1.30, 2.17], *p* < 0.001
			(B) ≥10 mg pred baseline: non SC	27/640 (4)	A vs. C	6.1% vs. 19.7%, *p* = 0.01	1.4 m vs. 3.4 m HR: 1.87 [1.43, 2.45], *p* < 0.001	2.2 m vs. 11.2 m HR: 2.38 [1.78, 3.19], *p* = 0.001
			(C) 0 to <10 mg pred baseline	557/640 (86)	B vs. C	22.2% vs. 19.7% *	4.6 m vs. 3.4 m HR: 0.77 [0.50, 1.19], *p* = 0.24	10.7 m vs. 11.2 m HR: 0.93 [0.59, 1.48], *p* = 0.77
Pinato (2020) [57]	HCC	Multiple	(A) ≥10 mg pred baseline	14/304 (5)	A vs. B + C	*p* = 0.62	6.7 m vs. 5.8 m, *p* = 0.37	10.4 m vs. 12.2 m, *p* = 0.48
			(B) ≥10 mg pred during ICI	64/304 (20)	B vs. A + C	*p* = 0.62	8.1 m vs. 10.7 m, *p* = 0.46	16.1 m vs. 11.7 m, *p* = 0.25
			(C) no GC at all	226/304 (75)	Among A + B: SC vs. non	SC: “more ICI refractory” *p* = 0.05	1.6 m vs. 8.8 m, *p* < 0.01	4.9 m vs. 15.4 m, *p* = 0.05
Umehara (2021) [58]	NSCLC	Nivo	(A) GC at baseline multiple indications	12/109 (11)	A vs. C	8% vs. 14%, *p* = 0.03	0.9 m vs. 3.3 m, *p* < 0.01	2.2 m vs. 11.9 m, *p* < 0.01
			(B) GC during ICI: irAE or no	19/109 (17) 14/109 (13)	B vs. C	36% vs. 14%, *p* = 0.02	3.6 m vs. 3.3 m, *p* = 0.23	12.5 m vs. 11.9 m, *p* = 0.72
			(C) no GC at all	64/109 (59)	Among B: irAE vs. no	47% vs. 21%, *p* = 0.13	5.1 m vs. 2.2 m, *p* = 0.17	13.5 m vs. 12.5 m, *p* = 0.30
Gaucher (2021) [59]	Multiple	Multiple	(A) concomitant GC: irAE	21/372 (6)	A + B vs. C	16.9% vs. 27.8%, *p* = 0.025	-	HR: 1.25 [0.91, 1.71], *p* = 0.16
			(B) concomitant GC: other indication	56/372 (15)	A vs. B + C	28.6% vs. 27.8%, *p* = 0.30	-	HR: 1.04 [0.56, 1.95], *p* = 0.90
			(C) no GC at all	295/372 (79)	B vs. A + C	12.5% vs. 27.8%, *p* = 0.008	-	HR: 1.34 [1.05, 2.03], *p* = 0.046

NSCLC: non-small cell lung cancer—HCC: hepatocellular carcinoma—ipi: ipilimumab—nivo: nivolumab—pred: prednisone—NS: not specified. GC: glucocorticoids—SC: supportive care—BM: brain metastases (+/−: present or not at baseline)—irAEs: immune-related adverse events—m: months—d: days. ORR: overall response rate—CI: confidence interval—PFS: progression-free survival—OS: overall survival—HR: hazard ratio—* *p*-value not available.

**Table 2 cancers-15-02276-t002:** Systematic reviews and meta-analyses on the impact of corticosteroids with ICI treatment on survival [42,60,61].

Author (Year)	Type of Cancer	ICI	GC Regimen	GC Indication	*n* Studies (*n* Patients)	PFS: HR, [95% CI]	OS: HR, [95% CI]
Petrelli (2020) [42]	Multiple	Multiple	Multiple	Overall	16 (4045)	1.34 [1.02, 1.76], *p* = 0.03	1.54 [1.24, 1.91], *p* = 0.0001
				SC	3 (836)	-	2.5 [1.41, 4.43], *p* < 0.01
				BM	3 (1164)	-	1.51 [1.22, 1.87], *p* < 0.01
				irAEs	9 (926)	-	1.08 [0.79, 1.49], *p* = 0.62
Zhang (2021) [60]	NSCLC	Multiple	Multiple	Overall	14 (5461)	1.69 [1.51, 2.04], *p* = 0.009	1.82 [1.51, 2.18], *p* = 0.003
				SC	NS	1.55 [1.26, 1.92] *	1.94 [1.57, 2.20] *
				BM	NS	1.56 [1.23, 1.97] *	1.62 [1.41, 1.86] *
Jessurun (2021) [61]	Multiple with BM	Multiple	Multiple	Overall BM	15 (1113)	2.00 [1.37, 2.91], *p* = 0.007	1.84 [1.22, 2.77], *p* = 0.007
				NSCLC BM	4 (505)	-	2.43 [0.38, 15.77] *
				Melanoma BM	NS	-	1.67 [1.49, 1.87] *

NSCLC: non-small cell lung cancer—pred: prednisone—NS: not specified. GC: glucocorticoids—SC: supportive care—BM: brain metastases—irAEs: immune-related adverse events. ORR: overall response rate—CI: confidence interval—PFS: progression-free survival—OS: overall survival—HR: hazard ratio—* *p*-value not available.

**Table 3 cancers-15-02276-t003:** Studies analyzing the impact of baseline and/or concomitant antibiotics on response rate and survival with ICI and/or chemotherapy or targeted therapy [26,68,69,70,71,72].

Author (Year)	Type of Cancer	Treatment	ATB Regimen	*n* Patients/Total (%)	Subgroup	ORR [CI 95%]	PFS [CI 95%]	OS [CI 95%]
Routy (2018) [26]	NSCLC RCC UC	ICI (multiple)	Within 2 m before or 1 m after ICI initiation	69/246 (28)	Overall	-	3.5 m vs. 4.1 m, *p* = 0.017	11.5 m vs. 20.6 m, *p* < 0.001
				37/140 (26)	NSCLC	-	3.5 m vs. 2.8 m, *p* = 0.57	8.3 m vs. 15.3 m, *p* = 0.001
				20/67 (30)	RCC	-	4.3 m vs. 7.4 m, *p* = 0.012	23.4 m vs. 27.9 m, *p* = 0.15
				12/42 (29)	UC	-	1.8 m vs. 4.3 m, *p* = 0.049	11.5 m vs. NR, *p* = 0.098
				68/239 (28)	Validation cohort	-	2.6 m vs. 3.6 m, *p* = 0.24	9.8 m vs. 21.9 m, *p* = 0.002
Derosa (2018) [68]	RCC NSCLC	ICI (multiple) +/− TT	Within 30 d before ICI initiation	16/121 (13)	RCC	13% vs. 26%, *p* < 0.01	1.9 m vs. 7.4 m HR: 3.1 [1.4, 6.9], *p* < 0.01	17.3 m vs. 30.6 m HR: 3.5 [1.1, 10.8], *p* = 0.03
				48/239 (20)	NSCLC	13% vs. 23%, *p* = 0.26	1.9 m vs. 3.8 m HR: 1.5 [1.0, 2.2], *p* = 0.03	7.9 m vs. 24.6 m HR: 4.4 [2.6, 7.7], *p* < 0.01
Pinato (2019) [69]	Multiple	ICI (multiple)	(A) within 30 d before ICI initiation	29/196 (15)	A: overall	8% vs. 43%, *p* < 0.01	-	2 m vs. 26 m HR: 3.4 [1.9, 6.1], *p* < 0.01
				6/107 (6)	A: NSCLC	-	-	2.5 m vs. 26 m HR: 9.3 [4.3, 19], *p* < 0.01
				17/38 (45)	A: melanoma	-	-	3.9 m vs. 14 m HR: 7.5 [1.7, 30.4], *p* < 0.001
			(B) concomitant	68/196 (35)	B	-	-	NR vs. 26 m HR: 0.9 [0.5, 1.4], *p* = 0.65
Tinsley (2019) [70]	Multiple	ICI (multiple)	Between 2 w before and 6 w after ICI initiation: single vs. cumulative course	92/291 (32)	Overall		3.1 m vs. 6.3 m HR: 1.40 [1.03, 1.92], *p* = 0.033	10.4 m vs. 21.7 m HR: 1.47 [1.04, 2.11], *p* = 0.033
				NS	Single course	-	3.7 m vs. 6.3 m HR: 1.32 [0.80, 2.20], *p* = 0.28	17.7 m vs. 21.7 m HR: 1.26 [0.82, 1.93], *p* = 0.29
				NS	Cumulative courses	-	2.8 m vs. 6.3 m HR: 2.63 [1.25, 6.13], *p* = 0.026	6.3 m vs. 21.7 m HR: 1.90 [1.18, 2.08], *p* = 0.009
Cortellini (2021) [71]	NSCLC TPS > 50%	Pembro (A) vs. CT (B)	Within 30 d before initiation	(A) 131/950 (14)	A	30.1% vs. 44.4% OR: 0.57 [0.37, 0.87], *p* = 0.01	4.8 m vs. 7.5 m HR: 1.29 [1.04, 1.59], *p* = 0.02	10.4 m vs. 17.2 m HR: 1.42 [1.13, 1.79], *p* = 0.002
				(B) 87/595 (15)	B	33.3% vs. 37.6%, *p* = 0.50	5.1 m vs. 5.9 m HR: 1.10 [0.86, 1.40], *p* = 0.42	13.2 m vs. 14.9 m HR: 1.23 [0.95, 1.61], *p* = 0.11
Cortellini (2021) [72]	NSCLC	CT + ICI 1st line	(A) within 30 d before ICI initiation	47/302 (16)	A: overall	42.6% vs. 57.4% OR: 0.83 [0.42, 1.64], *p* = 0.60	5.6 m vs. 6.3 m HR: 1.12 [0.76, 1.63], *p* = 0.56	11.2 m vs. 16.6 m HR: 1.42 [0.91, 2.22], *p* = 0.12
				17/302 (6)	A: ATB > 7 d	-	HR: 1.31 [0.73, 2.31] *	HR: 1.76 [0.83, 3.71] *
				20/302 (7)	A: ATB IV	-	HR: 1.67 [0.88, 3.17] *	HR: 1.44 [0.69, 3.09] *
				12/76 (16)	A: among TPS > 50%	-	7.0 m vs. 9.8 m HR: 1.48 [0.62, 3.53], *p* = 0.37	16.3 m vs. 25.9 m HR: 1.61 [0.57, 4.49], *p* = 0.36
			(B) concomitant	117/302 (39)	B	-	HR: 1.20 [0.89, 1.63], *p* = 0.22	HR: 1.29 [0.91, 1.84], *p* = 0.15

NSCLC: non-small cell lung cancer—RCC: renal cell carcinoma—UC: urothelial carcinoma—TPS: PD-L1 tumor proportion score. pembro: pembrolizumab—TT: targeted therapy—CT: chemotherapy—ATB: antibiotics—NR: not reached—IV: intravenous—m: months—w: weeks—d: days. ORR: overall response rate—CI: confidence interval—PFS: progression-free survival—OS: overall survival—HR: hazard ratio—* *p*-value not available.

**Table 4 cancers-15-02276-t004:** Systematic reviews and meta-analyses on the impact of antibiotics with ICI treatment on response rate and survival [74,75,76,77].

Author (Year)	Type of Cancer	ICI	ATB Regimen	Subgroup	*n* Studies (*n* Patients)	ORR: OR, [95% CI]	PFS: HR, [95% CI]	OS: HR, [95% CI]
Lurienne (2020) [74]	NSCLC	Multiple +/− CT or TT	Multiple	Overall	23 (2208)	-	1.47 [1.13, 1.90], *p* < 0.01	1.69 [1.25, 2.29], *p* < 0.01
				Within 90 d before ICI	4 (708)	-	1.56 [0.78, 3.13] *	2.49 [0.95, 6.51] *
				Within 60 d before ICI	3 (325)	-	2.00 [1.34, 2.99] *	2.94 [1.60, 5.40] *
				60 d before to 60 d after ICI initiation	12 (1624)	-	1.72 [1.30, 2.27] *	2.04 [1.49, 2.79] *
				Within 90 d before ICI and during ICI treatment	5 (645)	-	0.97 [0.44, 2.17] *	1.24 [0.56, 2.76] *
Xu (2020) [75]	Multiple	Multiple +/− CT or TT	Multiple	Overall	20 (4331)	-	1.53 [1.30, 1.79], *p* < 0.01	1.90 [1.55, 2.34], *p* < 0.01
				NSCLC	12 (1880)	-	1.39 [1.16, 1.67], *p* < 0.01	1.73 [1.26, 2.38], *p* < 0.01
				NSCLC: ATB within 6 m before ICI	3 (515)	-	-	1.81 [0.91, 3.63], *p* = 0.09
				NSCLC: ATB within 1 m before ICI or during ICI	7 (1365)	-	-	2.09 [1.31, 3.32], *p* = 0.002
Wu (2021) [76]	Multiple	Multiple +/− CT or TT	Multiple	Overall	44 (12492)	0.61 [0.42, 0.90], *p* = 0.01	1.18 [1.11, 1.25], *p* < 0.01	1.20 [1.15, 1.25], *p* < 0.01
				RCC	4 (367)	0.30 [0.14, 0.67], *p* < 0.01	1.29 [1.19, 1.40], *p* < 0.01	1.12 [1.01, 1.25], *p* = 0.028
				NSCLC	9 (1276)	0.84 [0.50, 1.42], *p* = 0.51	1.13 [1.04, 1.23], *p* < 0.01	1.26 [1.15, 1.38], *p* < 0.01
				Melanoma	2 (182)	0.37 [0.12, 1.10], *p* = 0.07	1.75 [1.34, 2.29], *p* < 0.01	1.36 [1.06, 1.75], *p* = 0.017
				ATB before ICI	8 (1060)	0.47 [0.32, 0.71], *p* < 0.01	1.23 [1.14, 1.32], *p* < 0.01	1.39 [1.26, 1.54], *p* < 0.01
				ATB before or after ICI within 1 m	9 (1010)	0.63 [0.32, 1.26], *p* = 0.19	1.16 [1.06, 1.26], *p* < 0.01	1.17 [1.10, 1.24], *p* < 0.01
Luo (2022) [77]	RCC	Multiple +/− TT	Multiple	Overall	6 (1104)	0.58 [0.41, 0.84] *	1.77 [1.25, 2.50] *	1.69 [1.34, 2.12] *
				60 d before to 60 d after ICI initiation	4 (NS)	-	1.86 [1.18, 2.95] *	1.66 [1.30, 2.11] *
				Within 90 d before ICI	2 (NS)	-	1.75 [0.40, 7.55] *	0.66 [0.13, 3.35] *

NSCLC: non-small cell lung cancer—RCC: renal cell carcinoma—CT: chemotherapy—TT: targeted therapy—ATB: antibiotics—NS: not specified—m: months—d: days. ORR: overall response rate—CI: confidence interval—PFS: progression-free survival—OS: overall survival—HR: hazard ratio—OR: odds ratio—* *p*-value not available.

**Table 5 cancers-15-02276-t005:** Studies analyzing the impact of proton pump inhibitors on response rate and survival with ICI and/or chemotherapy treatment [79,88,89,90,91].

Author (Year)	Type of Cancer	Treatment	PPI Regimen	*n* Patients/Total (%)	Subgroup	ORR [CI 95%]	PFS [CI 95%]	OS [CI 95%]
Hopkins (2020) [88]	UC	Atezo or CT (IMvigor210, 211)	Within 30 d before (A) or after (B) ICI initiation	286/896 (32)	Pooled atezo	OR: 0.51 [0.32, 0.82], *p* = 0.006	HR: 1.38 [1.18, 1.62], *p* < 0.001	HR: 1.52 [1.27, 1.83], *p* < 0.001
				185/464 (40)	CT	OR: 1.04 [0.64, 1.71], *p* = 0.2	HR: 1.11 [0.89, 1.37], *p* = 0.35	HR: 1.16 [0.93, 1.47], *p* = 0.2
				272	Atezo + PPI: A vs. B	-	HR: 0.71 [0.49, 1.03], *p* = 0.07	HR: 0.65 [0.44, 0.97], *p* = 0.033
Chalabi (2020) [79]	NSCLC	Atezo or CT (OAK, POPLAR)	Within 30 d before or after ICI initiation	234/757 (31)	Pooled atezo	-	1.9 m vs. 2.8 m HR: 1.30 [1.10, 1.53], *p* = 0.001	9.6 m vs. 14.5 m HR: 1.45 [1.20, 1.75], *p* < 0.001
				260/755 (34)	CT		3.5 m vs. 3.9 m HR: 1.04 [0.89, 1.22] *	9.1 m vs. 11.0 m HR: 1.17 [0.97, 1.40] *
				74/757 (10)	Pooled atezo: PPI + ATB	-	1.7 m vs. 2.8 m HR: 1.48 [1.16, 1.91] *	6.6 m vs. 14.1 m HR: 1.89 [1.42, 2.52] *
Stokes (2021) [89]	NSCLC (US veterans)	ICI (multiple) +/− CT	Within 90 d of ICI initiation	2159/3634 (59)	Overall	-	-	10 m vs. 10 m HR: 0.98 [0.90, 1.06], *p* = 0.59
Baek (2022) [90]	NSCLC	Multiple (L2+)	Within 30 d before ICI initiation (new users or not)	936/2963 (32)	Overall	-	-	5.1 m vs. 8.0 m HR: 1.28 [1.13, 1.46], *p* < 0.001
				168/2963 (6)	New PPI users	-	-	3.8 m vs. 8.4 m HR: 1.64 [1.25, 2.17], *p* < 0.001
Peng (2022) [91]	Multiple	Nivo or pembro +/− CT	Within 30 d before or after ICI initiation	89/233 (38)	Overall		HR: 1.05 [0.76, 1.45] *	HR: 1.22 [0.80, 1.86] *
				46/117 (39)	NSCLC	-	HR: 1.33 [0.86, 2.04] *	HR: 1.18 [0.79, 2.01] *

UC: urothelial carcinoma—NSCLC: non-small cell lung cancer—RCC: renal cell carcinoma. Atezo: atezolizumab—nivo: nivolumab—pembro: pembrolizumab—L2+: second-line or later treatment. CT: chemotherapy—PPI: proton pump inhibitors—ATB: antibiotics—m: months—d: days. ORR: overall response rate—CI: confidence interval—PFS: progression-free survival—OS: overall survival—OR: odds ratio—HR: hazard ratio—* *p*-value not available.

**Table 6 cancers-15-02276-t006:** Systematic reviews and meta-analyses on the impact of proton pump inhibitors on ICI treatment survival [93,94,95,96].

Author (Year)	Type of Cancer	ICI	PPI Regimen	Subgroup	*n* Studies (*n* Patients)	PFS: HR, [95% CI]	OS: HR, [95% CI]
Li (2020) [93]	Multiple	Multiple	Prior or within	Overall	7 (1482)	0.90 [0.66, 1.23], *p* = 0.51	1.05 [0.79, 1.40], *p* = 0.73
				NSCLC	4 (NS)	1.17 [1.05, 1.31], *p* = 0.006	1.24 [1.00, 1.55], *p* = 0.05
				Melanoma	2 (NS)	0.50 [0.28, 0.91], *p* = 0.02	0.67 [0.30, 1.52], *p* = 0.34
Liu (2022) [94]	Multiple	Multiple +/− TT	Prior or within	Overall	17 (9978)	1.19 [0.98, 1.44] *	1.29 [1.10, 1.50] *
				30 d before and/or after ICI initiation	5 (NS)	1.23 [1.06, 1.43], *p* = 0.007	1.38 [1.18, 1.62], *p* < 0.001
				Any time after ICI initiation	7 (NS)	0.72 [0.40, 1.28], *p* = 0.18	1.27 [1.01, 1.59], *p* = 0.038
				NSCLC	6 (NS)	1.27 [1.10, 1.47], *p* = 0.001	1.19 [0.92, 1.54], *p* = 0.18
				Melanoma	2 (NS)	0.48 [0.25, 0.90], *p* = 0.023	0.70 [0.31, 1.56], *p* = 0.38
Chen (2022) [95]	Multiple	Multiple	Prior or within	Overall	33 (15,957)	1.30 [1.17, 1.46], *p* < 0.001	1.31 [1.19, 1.44], *p* < 0.001
				At baseline	3 (2194)	1.29 [1.15, 1.44], *p* < 0.001	1.43 [1.21, 1.69], *p* < 0.001
				Within 60 d before ICI initiation	20 (7742)	1.33 [1.20, 1.48], *p* < 0.001	1.35 [1.22, 1.51], *p* < 0.001
				After ICI initiation	12 (>4900)	1.19 [0.65, 2.17], *p* = 0.58	1.18 [0.98, 1.41], *p* = 0.09
				NSCLC	13 (9200)	1.33 [1.17, 1.51], *p* < 0.001	1.33 [1.15, 1.54], *p* < 0.001
				RCC	6 (433)	1.11 [0.89, 1.38], *p* = 0.37	1.01 [0.77, 1.33], *p* = 0.92
Dar (2022) [96]	NSCLC	Multiple +/− TT	NS	Overall	4 (2940)	1.31 [1.17, 1.47], *p* < 0.01	1.46 [1.27, 1.67], *p* < 0.01

NSCLC: non-small cell lung cancer—RCC: renal cell carcinoma—TT: targeted therapy—PPI: proton pump inhibitors—NS: not specified. PFS: progression-free survival—OS: overall survival—HR: hazard ratio—CI: confidence interval—m: months—d: days—* *p*-value not available.

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
