# Peer review of "Comedications with Immune Checkpoint Inhibitors: Involvement of the Microbiota, Impact on Efficacy and Practical Implications"

_cancers, 2023, doi:10.3390/cancers15082276_

Round 1
Reviewer 1 Report
The paper is focuses on the Comedications with immune checkpoint inhibitors: A review on microbiota’s involvement, impact on efficacy and practical implications. Some sections must be further developed before any considerations for publications. I suggest the following:
Shape suggestions
Tables must be written in black, not in grey (difficult to read).
Content suggestions
It must be added a 2nd section of Methodology of literature search/selection or a similar title. It would therefore be advisable to present the methodology for selecting bibliographic resources (databases used and the reason for choosing those data basis, types of documents included, filtering results, inclusion/exclusion criteria for manuscripts: language, key words, duplicates, etc.). Moreover, have you searched (graphically) the impact of the topic on the general literature? (Providing a figure would be relevant, as it will highlight in the best way the topic frequency in the literature)
L86. Subsection 3.2. must be corrected as 2.2. and must be better detailed – I suggest checking and referring to https://doi.org/10.3390/diagnostics11061090
Line 116 please discuss the impact of probiotic supplementation in other types of cancer. Please better present the immune mechanisms that connect the gut microbiota with the development of cancer.
Subsection 2.4. The influence of Akkermansia muciniphila as a Key Gut Bacterium should be better developed .
Line 256 please discuss the impact of immune suppression on development of cancer cells. Detail the particularities of microbiota in patients with immune suppression.
Line 259 explain the impact of certain antibiotics on the development of altered gut microbiota and production of metabolites that can increase colon cancer risk.
Conclusions section. What is the novelty of your paper? Please clear present this aspect, L 638 – as the clinician have as tool all the published literature, no needing for searching a Review, when there are hundreds of papers, with original results, in the topic.
Reviewer 2 Report
Authors have meticulously compiled the available information on the role of microbiota in immune checkpoint inhibitors efficacy, and the impact of comedications. The review is comprehensive and will serve as an excellent source of information on the topic. It provides extended information on the relation between the microbiota and the immune system, impact of gut microbiota on the efficacy of immune checkpoint inhibitors, corticosteroids, antibiotics, proton pump inhibitors, and other medications. I only have a few minor observations:
1. The work would benefit from close editing. Throughout the manuscript, there are paragraphs that are shorter than usual (1-2 sentences). Consider consolidating them.
2. Methodology: Which databases were used? How were the references for this review selected? What were the reference inclusion and exclusion criteria?
3. Add a schematic diagram to summarize the relation between the microbiota and the immune system.
4. Rewrite the conclusion section.
Round 2
Reviewer 1 Report
The authors responded to my requests.